# Troublesome Access: Non-Admission Experiences in the Competitive Finnish Higher Education

**Ulpukka Isopahkala-Bouret**

Department of Education, University of Turku, 20500 Turku, Finland; ulpukka.isopahkala-bouret@utu.fi

**Abstract:** In this study, I address policy aims to reconcile equality of opportunity and marketization by examining difficulties in access to Finnish higher education. Finnish higher education is largely funded by the state and has no tuition fees. However, new demands have arisen that align with market-driven policy. At the same time, the Finnish system is one of the most competitive systems in the Organization for Cooperation and Development (OECD), and around 70% of applicants do not gain admittance. The purpose of this study is to examine how prospective degree students who have applied without being allowed to start studying toward a degree respond to the loss of opportunity and position themselves in the higher education marketplace. The analysis is based on 50 online narratives. The results are elaborated into three exploratory story models: (1) 'Never give up on your dreams'; (2) 'Need to figure out a new plan'; and (3) 'You can't get everything you want in life'. The stories show that marketization of higher education affects the experiences and expectations of prospective students. Moreover, marketization offers opportunities differently for those who already have plenty of resources to compete for access to higher education and those who do not.

**Keywords:** student admission; selection; non-admission; marketization of higher education; equality of opportunity

## 1. Introduction

Policies promoting marketisation of higher education and accompanying transnational reforms are part of neoliberal thinking, which aims to transform higher education in line with a corporate-like model (e.g., Brown and Carasso 2013; Collins 2013; Marginson 2013). Accordingly, higher-education institutions (HEIs) work best if their actions are driven by customer demand and competition. Students as (paying) customers make more deliberate choices and increase opportunities by requesting greater variety and higher-quality provisions. However, evidence pertaining to whether marketization increases student choice and participation is contradictory (Callender and Dougherty 2018; Bowl 2018; Brown and Carasso 2013; McCaig 2010).

Finland is a particularly interesting country to compare from an international perspective. The system of Finnish higher education, which is composed of 14 research universities and 25 universities of applied sciences, has been largely funded by the state. There are no tuition fees, and egalitarian principles are expressed openly in student admission policies. At the same time, the Finnish system is one of the most competitive higher education systems in the Organization for Cooperation and Development (OECD) (OECD 2019). For example, in 2018, 74% of the annual number of applicants did not gain admittance, and the acceptance rate varied between 4% and 31% in different disciplines and fields of study (Vipunen 2018). The competition for admission is bureaucratically governed and administrated. However, new demands on higher education have arisen that align with market-driven policy (Rinne et al. 2014). Simultaneously, a series of student admission reforms have taken place (Haltia et al. 2019).

In what follows, I elaborate first on how the marketisation rationale (without introduction of tuition fees) and accompanying neo-liberalist reforms have emerged in Finnish higher education policies. Thereafter, I analyze how prospective students who fail to gain admittance to a competitive Finnish HEI position themselves in the national and international higher education marketplace.

## 2. Background of Policy Reforms and Marketization in Finnish Higher Education

The key idea behind student choice and marketization is that a wider variety of higher education provisions can better address demand; therefore, students can attain higher education that better accords with their particular background and interests (e.g., Brown and Carasso 2013). The development of such a market environment puts particular pressure on state-regulated higher-education systems (cf. Bloch et al. 2018). The governance of higher education as a public good provided at uniformly high-quality levels, as has been the case in Finland, prevents profound market reforms (Marginson 2013). Marketisation entails increasing heterogeneity and vertical differentiation through national regulation. The provision of Finnish higher education underwent dramatic massification reform in the 1990s when the most advanced part of upper-secondary vocational education was integrated into the higher education system by establishing polytechnic institutions. Around 10 years after they were established, Finnish polytechnics started to use academic symbols and titles and are now called universities of applied sciences (Isopahkala-Bouret 2018). Furthermore, recent developments point to further differentiation as institutions with a formally homogeneous status are encouraged to create distinct research and teaching profiles, presumably attracting different student populations.

Policy reforms that render HEIs more entrepreneurial and market-oriented cover a wide range of features, such as emphasizing institutional status rankings, increased competition for funding, and responsiveness to the demands of external stakeholders (e.g., Ball 2012; Brown and Carasso 2013; Marginson 2013). Ongoing policy changes have affected how the respective public institutions in Finland are defined and run, how they justify their activities and practices, and how they form relations with various social actors, including students (Jauhiainen et al. 2015; Rinne et al. 2014; Hölttä et al. 2010). Recently, the Finnish University Act (Finlex 2009) and the University of Applied Science Act (Finlex 2014) have increased the financial autonomy of Finnish institutions and strengthened the rationale for revenues in the governance of higher education. Legally, Finnish universities are foundations or corporations under public law. Universities of applied sciences operate as limited public companies. In many respects, reforms under the new legislation have strengthened demands for effectiveness, excellence, profit-seeking, and immediate benefit.

In a marketized system, demand for and supply of higher education are balanced through the price mechanism (Brown and Carasso 2013). However, marketization can be seen as a set of processes rather than an end state (Bowl 2018). Therefore, public higher education systems can exhibit aspects of market-like behavior, although they are not fully marketized. The Finnish government has already promoted marketization in international education, and students outside of the European Economic Area have paid tuition fees since 2017. In 2018, annual fees varied between 2100 and 18,000 euros, and three out of four students received a grant, which ranged from 10% to 100% of the tuition fee (MoEC 2018). Educational export is also seen as a promising avenue for additional revenue for Finnish higher education. Moreover, in the ongoing policy debate, some interest groups are promoting a cost-sharing model and tuition fees for all students at Finnish universities (e.g., Määttänen and Vihriälä 2018), although this is not the mainstream view.

Furthermore, a non-regulated, private course market has emerged alongside the public higher education system. Private enterprises have started to offer training and tutoring to prepare prospective students for entrance examinations at public universities (Kosunen and Haltia 2018). The economic threshold to enter some of these private preparatory courses constitutes a unique obstacle in terms of access to higher education. The preparatory courses require a personal economic investment (i.e., course fees) of up to several thousand euros for some of the courses in the most exclusive disciplines, such as medicine, law, and economics. Such private tutoring effectively provides advantages to wealthy

students (Kosunen 2018). In addition, even though there are no private universities in the current Finnish higher education system, for-profit universities governed from abroad (mostly from Baltic countries) have provided degree programs in Finland since the second decade of this century (Kosunen and Haltia 2018; Kosunen 2018). Their 'market share' is marginal, but they indicate a demand for alternative higher education providers.

## 3. Admission to Finnish Higher Education and Ongoing Reforms

Student admission policies and practices have resisted major marketization efforts that involve implementation of student fees. However, student selection is a mechanism through which Finnish universities can optimize their chances for competitive public funding. Formally, Finnish universities can create their own admission policies, but such decisions are based heavily on their performance negotiations with the ministry (MoEC 2016). Universities are paid according to the agreed target number of graduates; therefore, it is not beneficial for them to recruit more students than the number for which the institution is rewarded for by the state. In addition, the government's newest core funding scheme has criteria that reflect strengthening of marketization policies (MoEC 2019). Starting in January 2021, 30% (currently 19%) of universities' public funding will depend on accomplished degrees, 5% (currently 2%) on continuous learning (i.e., fee-based, non-degreed training provision), 4% (currently 2%) from graduate employment, and 3% (currently the same) on student feedback. Under the competitive funding models, universities may start acting strategically in order to be accountable for their student outcomes; for example, by recruiting first-time students who are likely to gain credit points, graduate in time, and find quality employment.

Furthermore, a series of student admission reforms has taken place since 2010. The common driver has been that the admission system should be simpler, more efficient, and increase the competitiveness of Finnish higher education in both the domestic and international student markets (MoEC 2016). The slow transition of young adults from secondary to tertiary education has been a specific target of reforms (Haltia et al. 2019). First, a joint internet-based application system (opintopolku.fi) was introduced in 2014 with uniform national regulations and handling of pooled admissions for all higher education institutions. The unified system is a way to provide comparative information about Finnish universities and universities of applied sciences to prospective students. In the joint application system, applicants can apply for a maximum of six higher education places in their preferred order, and they will obtain an offer to the highest target on the list for which they have sufficient results.

Another recent change in Finnish admission policies is that, since 2016, a quota for first-time students has been implemented to enhance their chances of gaining admission. Finnish student admissions in all universities and universities of applied sciences are based on *numerus clausus*, that is, a limited number of student places. The implementation of a first-time students' quota did not increase the overall number of study places. Furthermore, admission criteria are changing in 2020 (MoEC 2016). Admission to Finnish HEIs has been on the basis of the matriculation examination score, success on an HEI entrance examination, or a combination of the two. The matriculation examination is the only standardized national test in the Finnish school system. Entrance examinations usually test knowledge of material selected by an individual HEI, though some programs (e.g., teacher education) also include interviews, group discussions, or other tasks to evaluate an applicant's suitability. The reform will dramatically reduce the importance of entrance examinations and give more weight to the results of upper-secondary schooling.

The planning committee of the ongoing admission criteria reform quoted at length (MoEC 2016, pp. 53–56) the expert economist from the Finnish Institute for Economic Research, a research unit operating under the administrative domain of the Ministry of Finance (Pekkarinen and Sarvimäki 2016). The proponents of the ongoing reform argued that the grades of the Finnish matriculation examination, which is the only nationwide standardized test, are to be used as criteria in student selection to accomplish a better match between the applicants' choices and the choices of universities, as well as fairness of access. They grounded the argument on the 'two-sided matching markets' theory

of Nobel-prize winner Alvin E. Roth (Roth 2015). Arguably, the current system forces prospective students to be calculative and make tactical choices (MoEC 2016; Pekkarinen and Sarvimäki 2016). Applicants must compare a variety of admission criteria and speculate about which entry examination to attend, the acceptance rate in different study programs, what options other prospective students may choose, and the academic level of other applicants (i.e., how intensive the expected competition is).

Moreover, if prospective students put a lot of effort into preparing for the entry exams for option A and do not earn a place, they are 'punished' by not having enough time to prepare for the entry exams for plan B, for which they may have a higher probability of being admitted (Pekkarinen and Sarvimäki 2016). As a result, potential students may not apply to the place they would like to attend the most but to the one that they think offers the highest chance of admission. In a better system, potential degree students could express all their choices without making any unnecessary calculations in order of preference. Hence, failing to gain admission to option A would not diminish their chances of being admitted to their alternative options.

Another argument for reducing the role of entrance examinations is to resist the expansion of the private preparation course market, which has emerged alongside the public education system. These courses are not formal higher education courses, and the government's interest is to reduce demand for such courses. Therefore, HEIs are encouraged to abandon entrance examinations that require long-term preparation. However, it seems unlikely that private preparatory courses will disappear even if entry examinations are no longer used. Instead, they will probably be reformulated and targeted at students in upper-secondary schools who are preparing for the matriculation examination (Kosunen and Haltia 2018).

Ongoing student admission reform will change the selection mechanisms through which places are allocated to potential students. Nevertheless, the demand for higher education will continue to exceed the number of available study places in Finland, especially in the most sought-after institutions and disciplines, and only potential degree students with the highest grade average and/or success in entry examinations can really make choices. The newly appointed government[1] has announced that in addition to ongoing student admissions reform, a number of new study places must be created in the Finnish higher education sector (Valtioneuvosto 2019). The government has stated that degree education shall remain free of change. However, the agenda for expanding the overall number of university students is in tension with attempts over the last few years to slash the public costs of higher education in Finland. Therefore, the future will tell whether the intended expansion of study places in the Finnish system of higher education will involve further marketization, such as allowing new providers to enter the system, including some private and for-profit institutions, which would radically change the higher-education landscape.

## 4. Research Problem, Data, and Method

International research has established that social background is a central factor in student choice and determining who applies and who is admitted to higher education (Reay et al. 2005). Moreover, students from privileged backgrounds predominantly choose programs with high entrance qualification requirements, which, in turn, lead to more lucrative career paths (Börjesson and Broady 2016; Nori 2018). Also, in Finland, admissions procedures seem to lead to a somewhat socially biased student population in which those from more affluent backgrounds are overrepresented in higher education (Nori 2018; Isopahkala-Bouret et al. 2018). Home life and parental educational and career patterns, together with social, cultural, and economic capital, all deeply influence students' final grades and entry examination scores, motivation statements, and interview success. Such factors also influence

---

[1] Prime Minister Antti Rinne's Government was appointed on June 2019. It is formed by the Social Democratic Party, the Centre Party, the Greens, the Left Alliance and the Swedish People's Party of Finland.

the availability of support and resources needed for access to higher education. Less known, however, is how applicants from various backgrounds experience non-admission to higher education.

In this study, I approach equality of opportunity from the perspective of non-admission. The aim is to examine how prospective degree students who have applied to a competitive Finnish university without being allowed to start studying toward a degree respond to the loss of opportunity and position themselves in the national and international higher education marketplace. The specific research questions are as follows:

1. How do prospective students who have experienced non-admission to Finnish higher education position themselves in the competitive higher education marketplace?
2. How do they consider, if at all, marketized (tuition-based) higher education opportunities in their future prospects?
3. What kinds of story models can be constructed on the basis of these experiences and anticipations?

The analysis is based on 50 online narratives written in 2014–2018 by prospective degree students who applied to selective Finnish HEIs. These narratives were not sampled in any systematic way; the first 50 cases were obtained on the basis of convenience and availability. The narratives were published on various lifestyle blogging sites (*n* = 15) and public discussion forums (*n* = 35). The online publishing platforms were provided by Google (blogspot.com); Aller Media Corporation that hosts the biggest discussion forum in Finland (suomi24.fi); five media houses that publish magazines, websites, and social media content targeted for young women; one regional newspaper corporation; and universities. The length and content of the online narratives varied. The shortest ones were only half-page anecdotes, serving different functions in the ongoing online discussion (sharing similarities and supporting the non-admission experiences of the prior writer, giving advice, or questioning and mocking others' experiences and educational choices). The most elaborate and reflexive narratives were up to four pages long.

Online narratives can be considered 'authentic' data in the sense that the researcher has not initiated their production (Hakala and Vesa 2013). Moreover, people have shared sensitive, private experiences by using pseudonyms and revealing only limited information about their identity. This anonymity can influence the content of the online narratives. People share certain experiences and state specific opinions in a way that they would not do in direct face-to-face communication (Hakala and Vesa 2013). Therefore, online narratives can provide different information than would be possible with other data-production methods. Concerning non-admission experiences, it might be easier for an individual to reveal failure, disappointment, and hardship in anonymous narratives than in face-to-face research interviews.

In practice, the extraction of non-admission narratives from social media proceeded through the following steps. First, I selected keywords and phrases (e.g., 'I wasn't accepted to study'; 'I didn't gain access to university'; 'entry examinations'; 'university admission') and entered them into an internet search engine to identify discussions and blogs related to student admissions. Second, I browsed through the content of these sites to find more discussions about the topic. Third, I selected non-admission narratives from the overall mass of online discussions according to predefined criteria; only first-person stories sharing a specific non-admission experience were included. As a minimum, narratives needed to have a sequence of events (temporality), some evaluative elements (stating the point of a story), and some (speculative) consequences (Riessman 2008). The online narratives were originally published in Finnish. I translated the quotations used in this article by using both direct linguistic and sociocultural translation. Translation also served ethical aspects, because now the quotations cannot be traced directly back to the Internet.

The online narratives originated from what has happened—the application to higher education institution(s) and the unexpected experience of non-admission. At the same time, these narratives were open-ended, undecided, and directed toward the future. In their online narratives, people also wrote about their expectations regarding the upcoming gap year and further attempts to access higher

education. The accounts provided a view of uncertain but possible future scenarios (Uprichard 2011) and revealed how the prospective students position themselves in the educational market. Narratives of the future depend on the narratives of the past and present (Uprichard 2011). In this sense, they can be used to analyze how people perceive their possible educational futures in the context of their overall life histories, especially the various reasons why they want to obtain a higher-education degree.

I analyzed the online narratives by using an approach that resulted in emplotted story models (Polkinghorne 1995). First, I located common themes among the stories collected. Then, I synthesized and configured these recurrent themes into accounts that united and gave meaning to the data according to the purpose of research (Polkinghorne 1995, p. 15). As a result, I elaborated the prospective students' self-positioning and perceived (marketized) opportunities into three exploratory story models. These story models did not directly represent any individual's narrative. As such, the individual cases did not straightforwardly map onto these story models but had more variety and detail than what these models can entail. Still, as a rough estimate, one-half of the narratives fell into the Story Model I, one third into the Story Model II, and the remaining into the Story Model III. Furthermore, these story models aimed at critically interpreting the prevalence of marketization; therefore, I emphasized marketized higher education in future scenarios more than most individual narratives did. The narratives were prototypical in demonstrating how opportunities to recognize and 'use' higher-education markets are institutionally and socially bounded.

The limitations of data and methods used in this study yielded some concern. Foremost, the small number of individuals who participated in online discussions and write blogs cannot represent the experiences of the thousands of prospective students who face non-admission to selective Finnish HEIs each year. Furthermore, the anonymity of the online narratives meant that there was no way to gain any further information about the writers' background characteristics, such as socio-economic status, gender, and age, nor was not possible to ask for further information about the content or context of the non-admission experiences. However, I developed a method for analyzing and configuring story models to overcome these shortcomings. As a result, the focus was not on individuals' unique experiences but on elaborating culturally recognized models in terms of how prospective students can position themselves in the higher education market after non-admission experiences. Despite the limitations of the data, using the story-model approach, I was able to produce rich descriptions of how prospective students facing non-admission responded to their situation.

## 5. Results

As a result of narrative analysis, I constructed three distinctive story models. Each one described prospective degree students' experiences with non-admission to a selective Finnish HEI, though with different responses and future anticipation. The main themes and plotted elements that distinguished between the models are named as follows: preparation and anticipation for access; dealing with disappointment; reconsidering educational choice; and acknowledging the opportunities that the higher education 'market' can offer. The three story models are described in more detail below.

### 5.1. Story Model I: 'Never Give up on Your Dreams'

The prospective students in the first story model, 'Never give up on your dreams', position themselves as future students-to-be. Non-admission was seen as merely a barrier that they needed to overcome. As a background, they had a high attainment level in prior schooling, and they had completed general upper-secondary school with matriculation exams. Therefore, the choice of applying to higher education was expressed as a natural thing for them. Overall, around 85% of matriculated students in Finland apply to higher education (Vipunen 2018). The prospective students in the first story model expected to be selected because, for several months, they had prepared extensively for the entry exam and, in their opinion, had done well on the exam. In addition, their expectation of access to higher education was emphasized in the online narratives by recounting how parents, friends, and partners also expressed faith in their ability to gain admittance to higher education.

Non-admission was painfully experienced as shame, although it happens to many applicants every year. *"I am absolutely bothered, feeling that I've failed, and am ashamed to write this online post and tell about my own failure"* (blog 4). The prospective students representing the first story model experienced difficult emotions and self-doubt after they received the results of university rejection. They wondered what went wrong and why they did not get in. Nevertheless, their dedication to becoming a higher education student was stronger than their disappointment. They were able to move from feelings of desperation to a positive and productive study motivation, or at least that is how they represented the case in social media:

At the moment (when I got the message that I was rejected), it felt like the whole world collapsed For the following twelve hours, I was crying. I felt angry and disappointed. ( ... ) You can feel displeased with the situation and grieve, but rather soon you shall turn that emotion to your advantage and motivation. (blog 2).

> I may feel angry and frustrated to start again to study for the entrance examination next fall, but I feel ready to fight as well ( ... ). I'm actually really excited and motivated to further improve my level of competence. (blog 10).

Although it is not unusual in Finland that students apply several times, the decision of whether or not to reapply to a selective university requires contemplation. After gathering new information about the study options, listening to advice from other people, comparing alternatives, and calculating their chances, these potential students exclusively chose to reapply to their desired occupation or discipline at a selective institution. They approached their 'dream' job and favored lifestyle as requiring a particular higher education degree. In this case, marketization of higher education operated as a resource for the prospective students' dreams and future desires (Haywood et al. 2011).

> I reflected on different options and concluded that there's nothing else than to study (at a selective university) that I want to do in this world right now. I didn't want to apply for the university of applied sciences or in any other domain. I decided to reapply one more time. (blog 2).

> It's difficult to imagine that I would be happy if I start studying something else ( ... ). I know what I want and that I would be good at that (profession). (discussion 16).

These prospective students in the first story model were eager to reflect upon possible shortcomings in their preparation and entry exam results. The national online application platform (opintopiste.fi) serves as an information source, including information about all the study options, admission criteria, and admission results from previous years. Furthermore, the prospective students were considering concrete methods to improve exam performance in the second round of applications. Hence, much of the online discussion was about how to engage in self-improvement activities during their gap year to improve their chances of admission. This included taking part in preparatory courses, attaining university courses at the Finnish Open University, working as a trainee in a relevant domain, or going abroad to study languages.

> Put effort into studying something outside of university. There are many alternatives. If in your domain the admission is based on the results of the matriculation exam, retake the test, and improve your grade average. That will improve your chances for admission. You can also take courses at Open University. ( ... ) I spent my two years off at work, both abroad and in Finland, and studied for the entry exams. ( ... ) I also attended a preparatory training. (blog 3).

The desperate desire to be admitted to Finnish public higher education after reapplication has opened up new 'markets' for non-degreed, preparatory higher education courses. Uncertainty and the risk of another failure in the admission process were the fuel for such demand. In a public

administrative system with strictly meritocratic selection, one cannot 'buy' access. However, it is possible to invest in (educational) resources that may facilitate access. Preparatory training does not guarantee access, but it provides hope and support (Kosunen and Haltia 2018).

> Preparatory training. If at all possible, participate! I was able to attend one (before reapplying), and I experienced it as really useful. ( . . . ) I was working part-time as little as possible, one or two shifts per week. But, of course, it would have been better to not work at all. (blog 5).

Prospective degree students in the first story model were privileged in the sense that they could afford private tutoring and extra courses as well as other productive gap-year activities. In addition, they had social networks to support their efforts. Intensive periods of preparation for reapplication required absences from work or reduced working hours. Some had savings from prior employment, but others used alternative sources of income (e.g., support from parents or a spouse). Moreover, the course fees did not constitute a barrier for them to attend private preparatory courses, which usually cost between 500 to 2000 euros but can go as high as 6000 euros in the most competitive study fields (Kosunen and Haltia 2018).

### 5.2. Story Model II: 'Need to Figure out a New Plan'

The non-admission experiences of prospective degree students in the second story model, 'Need to figure out a new plan', were characterized by changes in career plans and/or choice of higher-education institution. These prospective students had also prepared well for the entry exams, but after they received the results, they had to conclude that other applicants had done better than they had. Competition was so high that it was pointless to continue down the same path, because the risk of repeated disappointment was too high. The future should bring escape from that vicious circle.

> I've applied to such fields that were almost impossible for me to get admission. But then I realised that I must figure out a plan C, D and E if I want to move on from this vicious reapplication hassle. (discussion 22).

Prospective students in the second story model had applied to their primary study option and maybe to their plan B option, but they had not considered multiple alternatives before being faced with (several) non-admission(s). Among all Finnish applicants, only 24% to 35% apply to more than one study option at a time (Pekkarinen and Sarvimäki 2016). Applicants probably consider only one or two options because it is very difficult and even impossible (e.g., due to overlapping entry exam schedules) to prepare sufficiently for more than one exam at the same time. In the second story model, it was simply the result of planning for a future occupation and career by the prospective students; only one option seemed to be the right one at the time.

The consideration of alternative plans, that is, study options other than what the person prioritized initially, were often outside of the traditional university sector. They pointed to the institutions in the non-university higher-education sector, such as Finnish universities of applied sciences. Moreover, some prospective degree students in the second story model considered foreign universities as an alternative option. It is uncommon for Finnish students to study a full degree abroad, although Finns have been actively participating in student-exchange programs. However, during the last 10 years, the number of Finnish degree students studying abroad has doubled and is now equivalent to around 3% of higher education students in Finnish HEIs (EDUFI 2018). One of the biggest reasons for going abroad has been easier access to higher education in a foreign system.

> Why it is so difficult to start studying in Finland? In almost any other country, you will have easier access to higher education, because there is no similar student admission system as there is here in Finland. (discussion 35).

> I decided that finally I will apply to (another European country) to study medicine, because it is supposedly easier to get access there. (discussion 32).

The need to find an alternative to the competitive system of Finnish higher education opened up a new demand for a marketized, fee-based higher-education degrees. When access to public higher education became difficult, these alternatives became appealing if they were affordable or if student grants were available. Some providers used online discussion forums, where people shared their non-admission experiences as a marketing channel for selling private higher-education opportunities. They posted online advertisements for foreign tuition-based universities.

I would like to tell you about one more opportunity, especially if you are interested in international business, Asia and Australia. I work in a private business school (in Asia), and we are recruiting European students to increase the level of internationalisation of our student groups. We can provide student grants up to 100 per cent for Finnish students to cover tuition fees. (advertisement posted on a discussion forum).

*5.3. Story Model III: 'You Can't Get Everything You Want in Life'*

The non-admission experiences of prospective degree students in the third story model, 'You can't get everything you want in life', were characterized by the need for an escape from emotional pressure after non-admission, downplaying the desire to become a degree student, and a shortage of resources to take advantage of the educational marketplace.

I feel so shitty ( . . . ). I need to take a break for a year or two and think about it later whether I am still willing or able to reapply to university. (discussion 21).

Like prospective students in the first and second story models, these people experienced feelings of frustration, disappointment, sadness, and anger after non-admission. Moreover, they felt anxious and depressed about not knowing what to do next in life. In this story model, the solution was not to find renewed motivation and to reapply, nor did they consider looking for higher-education opportunities outside of Finnish public higher education because of binding family responsibilities or financial restrictions.

I don't have an objective to spend many years in the reapplication processes. I don't really know how I could finance several gap years in a clever way. I mean, how could I earn money and still have enough energy to fully study for the entrance examinations? (discussion 18).

I don't have any income, and it is difficult to try to get access to university, because much of what I can do depends on my husband's willingness to support my studies. (discussion 19).

Prospective students in the third story model expressed doubt about their chances of gaining entry even if they reapplied. Some had already applied several times. These people had not been low achievers at school, and one could expect them to get access to a selective higher education institution. However, they had come to a realization that working hard at school is not necessarily rewarded by admission, as promised by the meritocratic belief (Isopahkala-Bouret n.d.). Therefore, it was necessary for them to look for educational opportunities outside of higher education. They presented the scaling down of educational aspirations as a rational, pragmatic, and realistic alternative.

I was smart and able to draw a conclusion (after non-admission): if I can't get a study place (at a selective university), I may not have been able to succeed while working in that professional domain either. Well, I had good grades in general upper-secondary school and at the matriculation test, so I wasn't bad at school. Anyway, I studied for another diploma. (discussion 3).

For those who decided to acquire vocational qualifications, there are opportunities in the Finnish labor market, although they differ from the future scenarios of university graduates. Around 70% of those with vocational diploma will find employment, compared with 86% of those with higher

education degrees (Statistics Finland 2019). In 2018, the average monthly earnings of people with vocational qualifications was around 2900 euros, compared to graduates with a bachelor's degree and master's degree whose monthly earnings were around 3400 euros and 4000–4500 euros, respectively (Statistics Finland 2018). The labor market prospects without any qualifications are highly uncertain as most employers in Finland require educational credentials.

Prospective students in the third story model found alternative future trajectories in which they anchored their 'happiness' to things other than higher education admission and a degree. They shifted the discussion away from higher education applications and admissions to other life goals and to alternative ways of getting them. Instead of formal higher education, they still believed in self-orientated lifelong learning.

> You can succeed in life even without formal credentials. It has been proven many times. It happens all the time! Remember that formal studies take usually years. You can learn new skills and knowledge and take initiative without schooling. Go and seek answers to the questions you want to ask. Try and test. Thanks to the Internet, the highway to find information is open to everyone. (discussion 9).

> Live the life that you have, and keep your eyes open to different alternatives. ( … ) It might be that you'll never get to study your (desired discipline). When you accept that fact and start seeing how wonderful life is despite that, you can live happily and enjoy your future family and all the good moments you'll have together. You can't get everything in life, and you don't need to. (discussion 33).

Some examples of alternatives to higher education were the 'self-made person', who does not need formal schooling to learn what is important in life, and a future parent, who finds happiness in caring for family. Is there room for marketized higher education in this storyline? Would anything appeal to these people in the third story model once the door to degreed higher education is closed? International research on so-called 'challenger institutions' have shown that new private higher education provision is often targeted at people who have a demand for short-cycle programs, sub-degrees, commercial certificates, and licenses (Evans 2018), and such demand is on the increase.

## 6. Discussion

This narrative study about access difficulties contributes to a better understanding of what is involved in reconciling marketization and equality of opportunity in Finnish higher education. In this study, I distinguished the prospective students' different non-admission experiences according to three story models. The resulting story models were named: (1) 'Never give up on your dreams', (2) 'Need to figure out a new plan', and (3) 'You can't get everything you want in life'. From the original online narratives and the elaborated story models, I identified four key elements against which the prospective students positioned themselves in the competitive higher-education marketplace. These elements were named *preparation and anticipation for access*; *dealing with disappointment; reconsidering educational choice*; and *acknowledging (marketized) higher-education opportunities*.

The stories show that marketization of higher education affects the experiences and expectations of prospective students. Competition for admission to Finnish HEIs is bureaucratically administrated and on the basis of a strictly meritocratic selection system. However, the competitiveness of student admission has silently facilitated an emerging shadow education system and exogenous privatization of public higher education (Ball and Youdell 2008; Kosunen 2018). A large-scale non-admission phenomenon has created an increasing demand for marketized higher education. Private preparatory training and tutoring and public tuition-based courses provide support and ease the uncertainties of prospective students who are preparing to reapply (cf., Kosunen and Haltia 2018). Those people who determine that their chances are too low for admission to the Finnish system search for alternatives abroad (EDUFI 2018) or within private foreign-based provision offered in Finland outside of the official higher education system, where access is presumably easier.

Third, the analysis casts light on the third storyline, prospective students who withdrew from the competition in the higher education marketplace. They perceive their access opportunities to the meritocratic public system as well as the marketized (foreign) alternatives as too limited, and reposition themselves differently in relation to credentialist societal goals. These experiences highlight that having greater variety in higher education provision and adequate information about what admission to different kinds of study options requires does not necessarily bring about more choice. Instead, it can lead to a painful awareness of how limited one's ability to choose and gain access to higher education really is (e.g., Reay et al. 2005).

Moreover, the results of this study challenge ongoing policy reform as being sufficient in making access to Finnish higher education easier for the majority of applicants. On the contrary, it can make access more challenging for prospective students who do not have top grades from upper-secondary school and who have a non-traditional educational or social background, as well as for mature students (Haltia et al. 2019; Nori 2018). Marketization offers opportunities differently for those who already have plenty of resources to compete for access to higher education and those who do not (e.g., Bowl 2018; McCaig 2010). Also, in the Finnish case, it is the wealthiest group of prospective students who are able to take advantage of the services provided by private tutoring and tuition-based provision (Kosunen 2018). They can use their market advantage to increase their chances of winning the competition for access to the most prestigious sector of Finnish higher education or to foreign universities. The rest must live with what remains—non-degreed, marketized higher education that does not necessarily reward students for their investment.

**Funding:** This research received no external funding.

**Acknowledgments:** I would like to thank the reviewers and Nina Haltia for their helpful comments on an earlier version of this article.

**Conflicts of Interest:** There is no conflict of interest.

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
