# Peer review of "Troublesome Access: Non-Admission Experiences in the Competitive Finnish Higher Education"

_socsci, doi:10.3390/socsci8110302_

Round 1

Reviewer 1 Report

I am very impressed by the polish and professionalism of this piece. It is very well written, clear, and well referenced. It draws on an interesting qualitative data source  to explore an issue which as far as I am aware is quite novel- the plans of those who fail to gain access to a university place.

The paper addresses the paradox of a state-funded no-fees HE regime which however cannot provide enough study places for all qualified applicants. It starts with a very useful discussion of the Finnish HE system and policies, which I suspect few readers will be familiar with.  It questions whether those who fail in this very competitive system can be helped by marketisation - an interesting question. To start to answer it the author has analysed online narratives of  unsuccessful applicants and identifies three storylines; having another go, lowering sights and going for a lesser-rated HEI, or abandoning application to  HE. The methods  are well-explicated and their limitations acknowledged.

I have just a few suggestions for small additions:-

It would be useful to know the political makeup of the current government which is advocating the provision of extra study places p 4 I would like to see more information on the online sources of the narratives. What kind of blogs are these? And what are the fora? p5  Although as the author indicates this is clearly a small and non-representative  sample, it would be interesting to know what proportion of the narratives fell into each story model.  For story model 3 I think it would make it clearer what future is being embraced  if we were given some indication of what Labour market opportunities are available in Finland for those without degrees. Finally  P4 l 179  it should  be made clear that the research referred to comes from other countries.  

Author Response

I thank the reviewer for the comments on the earlier version of the manuscript. The questions asked pointed out issues that I had not properly explained.

In the following, I explain the revisions made. The modifications and additions, including new references, are marked with yellow colour in the manuscript.

Comment 1: It would be useful to know the political makeup of the current government which is advocating the provision of extra study places

Response 1: I have included an end note that informs that Antti Rinne’s government is formed by the Social Democratic Party, the Centre Party, the Greens, the Left Alliance and the Swedish People's Party of Finland.

Comment 2: p 4 I would like to see more information on the online sources of the narratives. What kind of blogs are these? And what are the fora?

Response 2: I have included the list of the providers of the online platforms on which the narratives were published. These can all be characterised as lifestyle blogs and discussion forums with a wide-range of discussion topics.

Comment 3: p5  Although as the author indicates this is clearly a small and non-representative  sample, it would be interesting to know what proportion of the narratives fell into each story model. 

Response 3: Although, as I stated, the individual cases could not straightforwardly map onto the Story Models, I have given a rough estimate on what proportion of the narratives fell into each model

Comment 4: For story model 3 I think it would make it clearer what future is being embraced  if we were given some indication of what Labour market opportunities are available in Finland for those without degrees.

Response 4: I have provided some comparison between the employment and earning prospects of people with vocational qualifications and those with a university degree in Finland.

Comment 5: Finally  P4 l 179  it should  be made clear that the research referred to comes from other countries.  

Response 5: I have modified the text accordingly.

Reviewer 2 Report

This is a very interesting paper that shines a light on an original aspect of marketisation within a national system based on publically funded HE that is espousedely meritocratic. This is concerned with the positional advantage enjoyed on those from wealthy backgrounds that can access privately funded tuition and preparation support to improve their chances of success in application to highly selective institutions. The research employs a suitable and relatively large scale data collection method that enables the author(s) to fully explore the impacts of and responses to failure.

The paper links well with the themes of the Journal special issue and supports a key theme identified by Bowl (2018), that marketisation tends to play out differently in different national and historical contexts through the application of policy levers unique to that national context and the specific ways that the tensions between expansion, quality and access are addressed. While the author(s) reiterate that the Finnish HE system doesn't feature tuition fees or overt competition between providers, yet the market effect occurs at a slightly earlier point, leaving institutions themselves to justify their behaviour on the basis of meritocracy.   

Author Response

Thank you for the review comments.